# Medical Students’ Perspectives on LGBTQI+ Healthcare and Education in Germany: Results of a Nationwide Online Survey

**DOI:** 10.3390/ijerph191610010

**Published:** 2022-08-13

**Authors:** Gerrit Brandt, Jule Stobrawe, Sophia Korte, Livia Prüll, Nora M. Laskowski, Georg Halbeisen, Georgios Paslakis

**Affiliations:** 1University Clinic for Psychosomatic Medicine and Psychotherapy, Medical Faculty, Campus East-Westphalia, Ruhr-University Bochum, Virchowstr. 65, 32312 Luebbecke, Germany; 2German Medical Students’ Association (BVMD e.V.), Standing Committee on Sexual and Reproductive Health and Rights including HIV and AIDS (SCORA), 10115 Berlin, Germany; 3Institute of Functional and Clinical Anatomy, University Medical Center Mainz, 55131 Mainz, Germany

**Keywords:** gay, healthcare, lesbian, medical education, training, trans*

## Abstract

The healthcare needs of lesbian, gay, bisexual, trans*, queer, and intersex (LGBTQI+) persons are often overlooked, prompting national and international calls to include diversity-related competencies into medical students’ training. However, LGBTQI+-focused healthcare education targets remain elusive, as surveys reveal considerable variability across national student populations. To generate empirical data and vocalize recommendations for medical education, we conducted the first nationwide online survey among 670 German medical students from 33 universities. Overall, most respondents reported low confidence regarding their medical training preparing them for LGBTQI+ patients, stated that LGBTQI+ themes were not covered during training, and agreed that the inclusion of such themes is urgently needed. In addition, we found gender and LGBTQI+ community member status to be key variables. Men scored lower in knowledge than women, while community members scored higher than non-community members. Similarly, community members reported higher comfort levels. Non-community men showed the highest levels of prejudice and efficacy beliefs, while at the same time had the lowest scores in contacts and the perceived importance of LGBTQI+-related teaching. Keeping subgroup differences in mind, we recommend that educational training should include LGBTQI+ healthcare aspects and address self-efficacy beliefs in future medical professionals to overcome LGBTQI+ healthcare disparities.

## 1. Introduction

Recent polls estimate that one in every 14 US Americans and central Europeans (around 7%) identifies as lesbian, gay, bisexual, trans*, queer, or intersex (LGBTQI+) [1,2]. Despite increasing international and national efforts towards respecting their fundamental rights [3], LGBTQI+ persons face continuing discrimination [4,5,6,7], including in healthcare settings [8,9,10]. LGBTQI+ persons report being denied treatment after disclosing their sexual orientation or gender identity [9,11]. Patient dissatisfaction and mistrust, in turn, may reinforce withholding important information from healthcare professionals [12], leading to misdiagnosis and inadequate medical treatment [13]. Ongoing discrimination in healthcare settings is particularly concerning given that LGBTQI+ persons are at an exceedingly higher risk for chronic physical and mental illness [14]. Heart disease, migraine, asthma, and chronic back pain are significantly more prevalent among LGBTQI+ persons [15]. Members of the LGBTQI+ community are less likely to attend cancer screenings and thus display increased rates of various cancers [16]. Similarly, LGBTQI+ persons are three to four times more likely to develop mental disorders than the general population [15]. Substance use and dependence are more prevalent among non-heterosexual than heterosexual persons [14,17]. Non-heterosexual persons also show higher rates of anxiety and depression disorders [18,19,20], and increased suicide rates [21]. Given the vulnerabilities of LGBTQI+ persons [22] and the potentially severe consequences of concealing one’s sexual orientation and gender identity in medical examinations [13], tackling discrimination in healthcare settings remains crucial.

Physicians’ prejudice against LGBTQI+ persons, a lack of knowledge of LGBTQI+ relevant healthcare needs, and low comfort levels and self-efficacy beliefs in working with LGBTQI+ patients are frequently noted as potential sources of healthcare discrimination [23,24], prompting national and international calls for including diversity-related competencies into medical students’ training [25,26]. In a systematic review, Morris et al. recently identified 13 LGBTQ+-focused educational programs for healthcare students and providers, concluding that specific interventions show promise for improving LGBTQI+-related attitudes, community- and health-related knowledge, and comfort levels in interacting with LGBTQI+ patients [24]. Cautioning against universal approaches, the authors further note the importance of accounting for medical students’ perspectives when planning educational interventions. Students’ awareness of health disparities and motivation for change may critically determine the success of bias-related trainings. Moreover, student perspectives may help identify relevant targets and preferred educational strategies among the plethora of available options [23], which appears crucial given the still low inclusion rates of LGBTQI+ content in medical curricula [27,28,29].

Thus far, students’ perspectives on LGBTQI+ healthcare and education remain barely explored, with an informal search of existing surveys revealing considerable variability between different student populations. Students surveyed at a US American medical school expressed high levels of comfort in interacting with LGBTQI+ patients, but felt unprepared by their formal education [30]. At other US American medical schools, medical students agreed on the importance of sexual history taking with LGBTQI+ patients, although they also expressed lower comfort levels when compared to other patient populations [31]. Medical students surveyed at a UK university expressed positive attitudes towards LGBTQI+ healthcare education as well as a need for improved training, with low confidence in using and clarifying unfamiliar sexual and gender terms emerging as a particular concern [32]. Similar results emerged in another UK survey, revealing low confidence levels in discussing patients’ gender identity independent of the number of years of medical training [33]. Surveys among German and Hungarian students further revealed that knowledge of and attitudes towards LGBTQI+-focused medical education may depend on student sociodemographic features, for example, their own gender [34] or their religiosity [35]. This suggests that LGBTQI+ healthcare education may require tailoring at both national and subgroup levels.

Given that information regarding medical students’ perspectives on LGBTQI+ healthcare and education remains sparse, we conducted the first nationwide online survey among medical students enrolled at several German medical faculties. Specifically, we assessed students’ LGBTQI+-related knowledge, prejudice, comfort levels, and their experience with and perspectives on LGBTQI+ healthcare and education. The goal of the present study was to generate empirical data and to vocalize recommendations regarding adaptations in medical education and training for future generations of physicians in Germany, to meet LGBTQI+ healthcare needs. In addition, we considered students’ gender (men vs. women) and LGBTQI+ community status (member of the LGBTQI+ community vs. non-member) as relevant variables in our analyses. Although recommendations derived from a national survey remain selective, we nevertheless expect that advancing this line of research contributes towards the overall goal of tackling discrimination and improving the situation of LGBTQI+ persons in healthcare settings.

## 2. Materials and Methods

### 2.1. Study Design, Setting, and Participants

We conducted an online cross-sectional survey of medical students’ perspectives on LGBTQI+-focused healthcare and education in Germany. All students enrolled in any medical faculty at a German university were eligible for the study. In exchange for their participation, survey respondents could opt in for a raffle covering fifteen EUR 25 vouchers for an online marketplace to purchase medical literature.

### 2.2. Recruitment

Respondents were recruited using a convenience sampling method, recruiting as many respondents as possible within a five-month window through emails from the working group Sexuality and Prevention of the German Medical Students’ Association (bvmd e.V.). The working group’s national coordination distributed links using a nationwide email distribution list reaching 1052 German medical students directly. The group further appealed to local representatives and sexual education project groups to distribute the survey via additional local email distribution lists. We also advertised the survey in medical lectures held at the Ruhr-University Bochum, Campus East-Westphalia. The recruitment period was from August to December 2021.

### 2.3. The Survey 

A 57-item survey was developed in a multistage process involving an expert panel of psychologists, physicians, experts in trans* health, and members of the Workgroup Sexuality and Prevention of the German Medical Students’ Association (Bundesvertretung der Medizinstudierenden in Deutschland (BVMD) e.V.). The panel generated an initial set of 117 items, which were evaluated for validity, objectivity, and comprehensibility, and subsequently reduced to a final set of 57 items (see Appendix A). 

Survey sections (detailed below) covered medical students’ (a) LGBTQI+-related knowledge, (b) prejudice, (c) amount of contact with LGBTQI+ persons in everyday life, (d) subjective comfort in contact with LGBTQI+ persons, (e) efficacy beliefs in adequately performing interviews and medical examinations in LGBTQI+ persons, (f) experiences with receiving teaching on LGBTQI+ topics, and (g) perceived importance of and personal desire for LGBTQI+-related teaching in medical curricula. A final section was included to assess sociodemographic data (gender, age, study major, place of study, current term) and LGBTQI+ community membership status. 

#### 2.3.1. Knowledge

Nineteen items were designed to assess LGBTQI+-related knowledge, including terminology (e.g., “intersex individuals are those whose biological sex cannot be clearly assigned”), prevalence (e.g., “the percentage of individuals in the general population who identify as not exclusively heterosexual is 3−10%”), aspects of social discrimination (e.g., “homosexual and bisexual men are above average victims of physical violence”), mental health (e.g., “LGBTQI+ persons show higher rates of anxiety disorders and depression than average”), and specific medical knowledge (e.g., “trans* women have a higher risk of suffering cerebral ischemia compared to biological women of the same age”). Care was taken to include a mixture of easier (i.e., general) and more difficult (i.e., specific) knowledge questions, as judged by the panel of experts developing the items. All questions were answered in a yes-or-no format.

#### 2.3.2. Prejudice

Nine items assessed prejudice towards LGBTQI+ persons. Exemplary items included, e.g., “trans* identity and pedophilia are inter-related”, or “most homosexuals encourage or entice others to adopt a homosexual life style”. All questions were answered in a yes-or-no format.

#### 2.3.3. Contact

Six items assessed contact with LGBTQI+ persons in everyday life (family, friends, work) in a yes-or-no format.

#### 2.3.4. Comfort

Fifteen items assessed the subjective degree of comfort while interviewing or examining LGBTQI+ persons using a 4-point Likert scale (yes, rather yes, rather no, no). Perceived comfort during sexual history taking and physical examinations was assessed with regards to patients identifying as heterosexual, homosexual, bisexual, and trans*. In each case, comfort in interacting with patients of the same or opposite gender was queried. Furthermore, we assessed the degree of comfort while being mentored by an educator (lecturer) who openly identified as homosexual, bisexual, or trans*. Finally, we also assessed the degree of comfort in the case of experiencing public displays of affection in heterosexual, homosexual/bisexual, and trans* couples.

#### 2.3.5. Efficacy Beliefs

Three items assessed the participants’ beliefs of how capable they feel of addressing the needs and special requirements of LGBTQI+ persons, for example, to take the sexual history of LGBTQI+ persons (e.g., “I am convinced that based on the knowledge I acquired so far, I am able to take a thorough sexual history of LGBTQI+ persons”). We also assessed the students’ general beliefs that LGBTQI+ individuals in Germany receive adequate medical care using three additional items on a 4-point Likert scale.

#### 2.3.6. Experienced Teaching

Three items assessed received teaching on LGBTQI+-related topics during their studies on a 4-point Likert scale (yes, rather yes, rather no, no). This was surveyed for the preclinical (semesters 1−4), clinical (semesters 5−10), and for the last, practical year (“praktisches Jahr”, PJ) of German medical school.

#### 2.3.7. Importance of Teaching

Furthermore, two items were included on the perceived importance of LGBTQI+ content as well as the personal desire for LGBTQI+-related teaching using a 4-point Likert scale (yes, rather yes, rather no, no).

### 2.4. Procedure

The online survey was hosted on a university-supplied, password- and SSL-secured webserver and implemented using the JavaScript-based open-source platform jsPsych [36]. Before starting the survey, participants were informed on the initial webpage about the study’s aims, their voluntary participation, data anonymity, data handling, and safety. All respondents provided informed, opt-in consent by ticking a consent box at the end of the page. Participants could opt in for a raffle covering fifteen EUR 25 vouchers for an online marketplace on medical literature by providing an email address at the end of the survey. Voucher winners were selected using an online random number generator. All email addresses were detached from survey data prior to analysis to ensure data anonymity. Only completed surveys were recorded.

### 2.5. Statistical Analysis

Overall survey responses were analyzed descriptively using means, standard deviations, and percentages. Univariate analysis of variance (ANOVA) was used to compare responses between respondent subgroups as defined by gender and LGBTQI+ community status. Tukey’s post hoc comparisons were used to follow up on significant interaction effects. Regression analyses were used to explore associations between potential targets of LGBTQI+-focused medical education (i.e., knowledge, prejudice, comfort, and efficacy beliefs) and students’ averaged teaching and private experiences. The significance level for all inferential analyses was set at *p* ≤ 0.05. No formal power analysis was conducted given the absence of prespecified inferential hypotheses. All analyses were performed with R in RStudio (2020; R version 3.5.6, R Foundation for Statistical Computing, Vienna, Austria; RStudio version 1.1.463, RStudio, PBC, Boston, MA, USA).

## 3. Results

### 3.1. Participants

Six hundred and eighty-three respondents (*n* = 683) completed the online survey. Thirteen individuals reported not to be studying medicine and were therefore excluded from analysis. Of the included 670 medical students, 276 self-identified as LGBTQI+ community members (196 female, 69 male, 11 diverse, M_age_ = 22.93, age range: 18 to 35 years), and 394 self-identified as non-LGBTQI+ (305 female, 89 male, 0 diverse, M_age_ = 23.93, age range: 18 to 50 years). The distribution of LGBTQI+ statuses did not differ across genders, according to a chi-square test of independence, χ^2^ (1, N = 626) = 0.84, *p* = 0.36. Participants indicated an average enrollment of 6.37 semesters, with *n* = 97 first year, *n* = 116 second year, *n* = 91 third year, *n* = 143 fourth term, *n* = 144 fifth year, and *n* = 79 higher than fifth year enrollment status. A visual examination of participant locations indicated nationwide distribution, with 33 universities included. 

### 3.2. Knowledge

Respondents correctly answered an average of 15.32 (SD = 1.91) out of 19 knowledge item (81%). Terminology items were correctly answered in 2.62 (SD = 0.63) out of three cases (87%); prevalence items were correctly answered in 2.34 (SD = 0.69) out of three cases (78%); social discrimination items were correctly answered in 3.09 (SD = 0.89) out of four cases (77%); mental health items received 2.87 (SD = 0.39) out of three correct answers (95%); and, finally, medical knowledge items were correctly answered in 4.4 (SD = 0.84) out of six cases (73%). When viewed individually, four items stood out as particularly difficult (Figure 1). These items addressed the percentage of non-heterosexual individuals in the general population (42% wrong answers), the discrimination of gay men (54% wrong answers), and specific medical knowledge on trans* individuals’ health (54% and 60% wrong answers).

### 3.3. Prejudice

LGBTQI+-related prejudice levels were low overall, with 0.37 (SD = 0.97) out of nine items answered affirmatively (4%). For example, 99% of respondents rejected a connection between trans* identity and pedophilia and disagreed with the statement that homosexual teachers may cause harm in the sexual development of children. Similarly, 99% of respondents negated that biological men who choose to dress as women are always homosexual. At the same time, 54 respondents (8%) agreed that due to the positive portrayal of homosexuality in the media, more young people would decide to become homosexual. Moreover, 8% agreed with the statement that due to gender-sensitive education in school, the number of trans* children would be rapidly increasing.

### 3.4. Contact

Consistent with low levels of prejudice, 82% of respondents indicated regular contact with at least one LGBTQI+ individual in everyday life. In total, 87% stated being friends with someone who openly identifies as homosexual or bisexual. Several respondents reported having a homosexual or bisexual individual in their family (21%). Compared to this, fewer medical students indicated contacts with trans* individuals; 21% of the respondents stated being friends with a trans* individual, while 3% reported having a close family member who openly identified as trans*.

### 3.5. Comfort

When asked about their perceived comfort in taking the sexual history from a heterosexual patient of the same sex, 93% of respondents gave an affirmative answer (Table 1). Somewhat fewer (81%) students responded affirmatively to do the same in a heterosexual patient of the opposite sex. Taking a sexual history from patients of the same sex identifying as homosexual or bisexual was felt to be comfortable by 91% of students, whereas 88% indicated this to be the case for homosexual or bisexual patients of the opposite sex. Medical students were least likely to feel comfortable interacting with trans* individuals, with 18% indicating they were not or somewhat not comfortable taking a sexual history from a trans* individual.

### 3.6. Efficacy Beliefs

Only 21% of respondents agreed that their medical education helped to identify and address the needs and special requirements of LGBTQI+ individuals within medical care. When asked if they felt confident taking a sexual history in LGBTQI+ individuals based on previous knowledge, 73% of medical students answered “no” or “rather no”, while 22% believed that LGBTQI+ individuals receive unrestricted medical care in Germany.

### 3.7. Experienced Teaching

Respondents reported only experiencing a little teaching on LGBTQI+ health aspects during medical school. The vast majority of respondents, 93%, reported “rather no” or “no” teaching at all on LGBTQI+-related topics during their preclinical studies. As many as 23% reported LGBTQI+ topics being addressed in the clinical semesters of medical school. Regarding LGBTQI+-related topics in the practical year of medical studies (“PJ”), 11% of respondents indicated “yes” or “rather yes”.

### 3.8. Teaching Importance

In total 91% of all participants considered LGBTQI+-related teaching in the medical curriculum as necessary and important, and 89% wished for more LGBTQI+-related teaching in medical curricula.

### 3.9. Subgroup Comparisons

In order to identify potential group differences among survey variables, respondent subgroups based on gender (men vs. women) and LGBTQI+ community status (community member vs. non- member) were compared using separate 2 (Gender) × 2 (Community Status) between-participants ANOVAs. Means, SDs, and inferential statistics are summarized in Table A1. N = 11 diverse respondents were excluded from analyses due to the small group size, but group means were nevertheless included in the table for the sake of completeness. Post hoc comparisons are summarized in Table A2.

Knowledge levels differed based on gender and community status, with men showing lower knowledge levels than women, *p* = 0.026, and members showing higher knowledge levels than non-members, *p* < 0.001. The interaction was not significant, *p* = 0.318.

Prejudice levels differed by gender and community status, and were qualified by a gender × community interaction, *p* < 001. Non-community men scored significantly higher than non-community women, *p* < 0.001, community women, *p* < 0.001, and community men, *p* < 0.001. Other comparisons were not significant, *p* > 0.43. 

Contact levels differed by community status, *p* < 0.001, and were qualified by a gender × community interaction, *p* = 0.01. Non-community men reported fewer contacts than non-community women, *p* = 0.006, community women, *p* < 0.001, and community men, *p* < 0.001. Non-community women reported fewer contacts than community women, *p* < 0.001, and community men, *p* < 0.001. Community men and community women did not differ in reported contact levels, *p* = 0.93.

Comfort levels differed only by community status, *p* < 0.001, but not by gender or their interaction, *p* > 0.58; community members reported higher comfort levels than non-members. 

Efficacy beliefs varied by gender, *p* < 0.001, and were qualified by a gender × community interaction, *p* < 0.001. The main effect of community status was not significant, *p* = 0.10. Post hoc comparisons revealed that non-community men scored higher than non-community women, *p* < 0.001, community women, *p* < 0.001, and community men, *p* = 0.001. All other comparisons were not significant, *p* > 0.99.

Reported levels of experienced teaching did not differ based on gender, community status, or their interaction, all *p* > 0.26. The perceived importance of teaching, however, differed by gender and community status, and were qualified by a gender × community interaction, *p* < 0.001. Non-community men reported lower levels of perceived importance than non-community women, *p* < 0.001, community women, *p* < 0.001, and community men, *p* < 0.001. Non-community women also scored lower than community women, *p* = 0.003. Differences between community men and women, and between non-community women and community men were not significant, *p* > 0.06. 

### 3.10. Predictive Analyses

We additionally explored predicting potential targets of LGBTQI+-focused medical education (i.e., knowledge, prejudice, comfort, and efficacy beliefs) from students’ averaged teaching experiences as opposed to private experiences (i.e., previous contact), and self-interest (i.e., LGBTQI+ community member vs. non-member). This could hint at the current state of facultative inclusion of LGBTQI+ healthcare in medical education and highlight areas of existing teaching efforts in need of further improvement.

Controlling for gender differences (1 = woman, 0 = man; participants identifying as “diverse” were again excluded due to low sample size) and differences in study duration (i.e., number of terms), a multiple linear regression showed that previous contact with LGBTQI+ community members (sum score) and community status (1 = LGBTQI+ community member, 0 = non-member) predicted increased knowledge levels, β = 0.16, *p* < 0.001, and β = 0.16, *p* < 0.001, respectively. Instead, increased teaching experiences predicted decreased knowledge levels, β = −0.09, *p* = 0.04, suggesting a potential mismatch between educational and state-of-the-art content. Female gender and study duration (expectedly) were positively associated with knowledge levels, β = 0.07, *p* = 0.04, and β = 0.15, *p* < 0.001, respectively.

Prejudice levels decreased with previous contact, β = −0.12, *p* = 0.006, and with community status, β = −0.11, *p* = 0.01, but were unaffected by teaching experiences, β = 0.06, *p* = 0.10. Female gender was negatively associated with prejudice levels, β = −0.14, *p* < 0.001, whereas study duration was not linked to prejudice levels, β = −0.08, *p* = 0.05.

Including comfort in sexual history taking and medical examination in heterosexual patients as an additional control, β = 0.83, *p* < 0.001, mean comfort levels in history taking and examination of LGBTQI+ patients were predicted by community status, β = 0.11, *p* < 0.001, and previous contact, β = 0.09, *p* < 0.001. Teaching experiences showed no effect, β = −0.003, *p* = 0.87. Female gender was not associated with comfort levels, β = 0.01, *p* = 0.57, whereas study duration was negatively associated with comfort levels, β = −0.05, *p* = 0.02.

However, efficacy belief levels increased with teaching experiences, β = 0.35, *p* < 0.001, and were not dependent on community status, β = −0.05, *p* = 0.24, and previous contact, β = −0.06, *p* = 0.12. Female gender was negatively associated with efficacy beliefs, β = −0.13, *p* < 0.001, whereas study duration had no effect, β = −0.01, *p* = 0.82. Thus, while students’ beliefs regarding their studies preparing them to address LGBTQI+ healthcare needs increased with teaching experience, knowledge, prejudice, and comfort levels primarily depended on private contacts and self-interest. 

## 4. Discussion

Ongoing discrimination in healthcare settings against LGBTQI+ persons highlights the need to expand medical training towards LGBTQI+-related healthcare needs. However, potential targets of healthcare education remain unclear, as previous surveys reveal considerable variability across different student populations, suggesting that LGBTQI+ healthcare education may require tailoring at both national and subgroup levels. To generate empirical data and to vocalize recommendations for advancing medical education in Germany, we conducted the first nationwide online survey among 670 medical students enrolled at 33 German universities. The survey revealed overall high rates of LGBTQI+-related general knowledge, low rates of prejudice, high rates of contacts to LGBTQI+ individuals, and high rates of subjective comfort in contact with community members in medical settings. However, most respondents also reported low confidence regarding their medical training sufficiently preparing them for interactions with LGBTQI+ members, found that LGBTQI+ persons do not receive unrestricted medical care, that LGBTQI+ themes were not covered as part of their training, and agreed that the inclusion of such themes is urgently needed. To the best of our knowledge, this is the first large-scale, nationwide survey of medical students’ perspectives on LGBTQI+ healthcare in Germany. 

The overall findings suggest a need for improving medical students’ training and boosting their efficacy beliefs to deal with LGBTQI+ patients’ healthcare needs. Indeed, our predictive analyses showed that teaching experiences and general study duration were unrelated to most potential targets of LGBTQI+-focused medical education. Knowledge levels even decreased with specific teaching experiences (although they increased with study duration), which could be explained by a mismatch between educational and state-of-the-art content. Instead, potential targets were found to benefit from previous contact and community status, which we believe indicates an overall low level of facultative inclusion of LGBTQI+ healthcare in current medical education.

Group-based comparisons further suggest considering the recipients’ demographics when tailoring educational efforts. We found that medical students who were non-community members showed lower levels of knowledge, higher levels of prejudice, lower contact rates, and expressed less comfort in dealing with LGBTQI+ patients. Moreover, students who identified as men, and non-community men especially, showed the lowest rates of knowledge, comfort, and contacts, and expressed the lowest perceived importance for including LGBTQI+-related teaching contents, while also exhibiting the highest levels of prejudice and efficacy beliefs. The pattern could be explained by lower awareness of LGBTQI+ healthcare issues, due to lowered exposure, inflating subjective confidence levels (for a similar pattern among general practitioners, see [37]). 

We would argue that the disparity between knowledge and efficacy beliefs in non-community men shows a further need for improving awareness regarding LGBTQI+ patients’ healthcare needs and struggles. After all, a lack of awareness, and resulting lack of interest, could explain why men in our sample advocated the least for including LGBTQI+ health in medical training. Indeed, there have been numerous calls to incorporate more diverse perspectives in medical education systematically [25,38,39,40], including in medical textbooks [41]. However, awareness towards LGBTQI+ healthcare could also improve from increased workforce diversity among medical students and professionals. Although increased representation of LGBTQI+ and other social groups among the medical workforce remains an elusive goal [39,42], the positive side effects of increased contact on student knowledge, prejudice, and comfort levels, as evidenced in our findings, suggest that increased representation is urgently needed. 

Our findings also corroborate the impression of US American, UK, and German single center surveys of medical students’ perceptions, which, given the cultural similarities, could suggest a common core of overarching educational targets. Overall, these surveys suggest positive attitudes and high levels of comfort in general contact, but low efficacy beliefs and a lack of preparation in dealing with LGBTQI+ patients [30,32,33,34]. Still, local disparities highlight a continuing need for considering each student population separately. Some studies identified discomfort with sexual history taking of LGBTQI+ patients as a particular area of concern among US American and Hungarian medical students [31,35], which were not observed among our survey’s respondents. Future research will be required to determine if such patterns originate from differences in curricula, cultural norms, or sample composition.

Taken together, the remaining differences in survey findings illustrate that educators should consider consulting pre-intervention data before developing educational training, which would allow materials to be tailored toward student-specific needs. Morris et al. [24] recently reviewed and identified effective training to improve medical students’ knowledge, attitudes, and comfort regarding LGBTQI+ healthcare needs. Programs designed to increase student knowledge through lectures or presentations were generally effective, even with only a single session (see also [43,44]). Student journal clubs on LGBTQI+ health represent another viable and participatory option for improving student knowledge [45]. Comfort and belief in being able to address LGBTQI+ patients improved through experiential learning sessions (e.g., interview exercises, role play), which newer studies corroborate further [46,47,48]. Though intergroup contact effectively promoted tolerance toward LGBTQI+ patients, attitudes were more difficult to change. Thus, a cornucopia of evaluated interventions could be used to tailor adequate interventions if combined with the identified student-specific needs.

### Limitations

Although conducted with a large sample of medical students from a range of German universities and addressing the role of respondent characteristics in subgroup comparisons, we cannot exclude that self-selection biases may have occurred in our study. About 30% of respondents self-identified as LGBTQI+ community members compared to prevalence estimates of 7.4% in Germany [2]. We suspect that the distribution mode may have promoted LGBTQI+ participation as invitation recipients of the particular student groups were likely familiar and already engaged with the subject. As a result, our findings could overestimate the medical student population’s knowledge and positive attitudes. However, as even individuals familiar with the subject expressed low belief in their training to prepare them for encountering LGBTQI+ patients in an adequate manner, calls for addressing student comfort and efficacy beliefs are still warranted.

We must further note potential limitations regarding our survey. Although developed by an expert panel to ensure content validity, the survey has not been evaluated in terms of its internal structure and psychometric properties. For example, we assessed LGBTQI+-related knowledge on a general level rather than for each subgroup individually, which previous surveys suggest may have revealed lower knowledge levels among specific subject areas (e.g., trans* health, [49]). Further psychometric evaluation is therefore required. As well as the structure of student knowledge, such future research should address different dimensions of prejudice, comfort, and efficacy beliefs, and evaluate their relevance for predicting student interactions with LGBTQI+ patients. Indeed, Morris et al. [24] noted that measures used in evaluating educational interventions have been rarely examined in terms of their consequences for behavior, which should be remedied in future research. 

Finally, designing educational programs not only requires considering students’ educational needs, but also considering barriers and difficulties encountered in practice. Thus far, practitioners’ perspectives on LGBTQI+ health remain rarely explored [46], and systematic comparisons to student perspectives are sought after. Ultimately, comparing student and practitioner perspectives would provide invaluable insights into designing curricula that best address future professionals’ needs.

## 5. Conclusions

Medical students in Germany agree on the importance of LGBTQI+-related healthcare education but require further training. We found gender and LGBTQI+ community member status to be key variables within the context of LGBTQI+-related medical training. Our results reveal that, overall, men had lower scores in knowledge than women, while community members had higher scores than non-community members. Similarly, community members reported higher scores in comfort levels. Finally, non-community men scored higher in prejudice, lower in contacts, higher in self-efficacy beliefs, and lower in the perceived importance of teaching than all other groups (non-community women, community men, community women). With these findings in mind, we recommend that educational training should include LGBTQI+ healthcare aspects and address self-efficacy beliefs in future medical professionals to overcome LGBTQI+ healthcare disparities.

## Figures and Tables

**Figure 1 ijerph-19-10010-f001:**
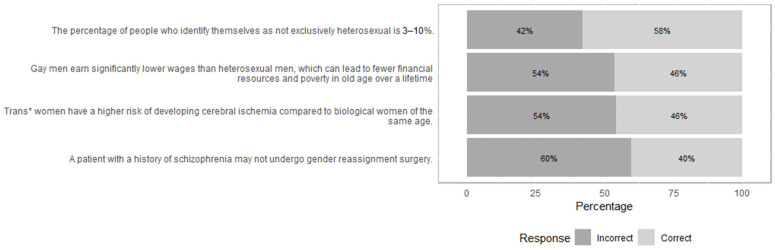
Selection of difficult items from knowledge assessment.

**Table 1 ijerph-19-10010-t001:** Relative response frequencies (%) from comfort assessment regarding sexual history taking and physical examination.

Group	Sex	Sexual History Taking		Physical Examination	
		Yes/Rather Yes	No/Rather No	Yes/Rather Yes	No/Rather No
Heterosexual	Same	93%	7%	97%	3%
	Other	81%	19%	89%	11%
Homo-/bisexual	Same	91%	9%	96%	4%
	Other	88%	22%	94%	6%
Trans*	-	82%	18%	88%	12%

## Data Availability

The data presented in this study are available upon reasonable request from the corresponding author.

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
