# Peer review of "Medical Students’ Perspectives on LGBTQI+ Healthcare and Education in Germany: Results of a Nationwide Online Survey"

_ijerph, 2022, doi:10.3390/ijerph191610010_

Round 1

Reviewer 1 Report

This is a very well thought-through study which is comparable to some others done in Europe, USA, Australia.  It's importance is in evaluating the situation for LGBTQI+ patients in Germany in terms of medical student training and education.  The importance if therefore in raising the profile of this issue such that there may then be opportunities to improve medical education provision and therefore care of this patient group in the future.

Line 16 in abstract should say "overlooked" not "overseen" - this is a vitally important change of semantics which it is essential to ensure correct meaning of the paper, not least because this is the abstract. I cannot find any other suggested edits

Reviewer 2 Report

I reviewed the manuscript “Medical Students’ Perspective on LGBTIQ+ Health Care Education in Germany” for the IJERPH.

The research topic is interesting, but the approach is punctual and generalizations are limited.

Title: Please make your title more accurate as far as findings are concerned and may maximize the chances of reach and citation; however, its main finding remains unclear. The use of questions is also interesting for the title.

Abstract: Suffers from the same problem as the title. I highlight the following problems:

1. It looks like it was written for professionals in the research area, and if so, it's less likely to be read. Understand your summary as a continuation of the title. The originality must be very clear in the abstract, and its conclusion must necessarily respond to the objective, respecting the adopted method; Structure in objectives, methods, results and main implications. The results must be highlighted

Results should present the key results, and not be tied to methodological details. In conclusion rather than repeating the results, authors should focus on answering the objectives thinking about the PRACTICAL IMPLICATIONS of their findings. In other words, why is your text interesting? What should an “average/non-expert” reader learn from your text? Highlight this in the conclusion.

Use the first person for this (we suggest that / we perceive that...). It is not possible to see the authors in the manuscript and this is a little disappointing.

Introduction:

The Introduction contains primary errors and fails to state the rationale for the research. It only limits itself to pointing out what other authors have already found, without differentiating, or pointing out where it is advancing.

In general, it needs to contextualize the theme nationally and internationally, there is no highlight for the knowledge gap reinforced by the literature.

It must necessarily POINTING AND PRESENTING the knowledge gap that it came to fill (HIGHLY RECOMMENDED); it is not enough to say that it exists. Remember that the gap must be auditable, so it is necessary to carry out a systematic search or point to a current (integrative or systematic) review;

To say that “There is a gap in knowledge...” is vague.

Method

There is a lot of information needed to analyze the data. Authors should follow the strobe checklist to guide the method: https://www.strobe-statement.org/checklists/

Data analysis is unclear and does not respect the principles of manuscripts that employ quantitative analysis.

The same applies to the results and tables, which must make clear the tests used.

What are the inclusion and exclusion criteria?

What is the origin of the seeds? How was it publicized? Was there dissemination among peers through social networks?

What are the allocation criteria? Was there a sample calculation? Was there a power test of the sample?

What are the platform's security criteria?

And what about the questionnaire? Was there validation? What is the process of creating it? Was there a pre-test? How and by whom?

Discussion:

The authors repeat some results in the discussion rather than explaining their findings. The results seem to continue and this is an error. You should focus on explaining/validating your original findings. What should the average reader take away from your discussion? Do not write to the specialist as he is able to understand your text.

Don't focus on just pointing out what the literature brings and justifying the similarities. For what reasons do they exist? Why are there differences? Explain, and validate your findings.

Remember to use arguments that validate your results.

This will require a readaptation of the discussion.

Study limitations must be DISCUSSED. How were the limitations circumvented so as not to influence the results?

Conclusion

The conclusion is very generic and similar to other sections. There is repetition. It must answer the objectives, be written in the first person and bring THE RESEARCHER'S VIEW ON THE FINDINGS. The conclusion is personal; that is, it is the look of you of the findings. Think about what your work changes, and why. It should not be focused on technical terms but on the implications.

Use direct, blunt language to explain what your findings were.

Author Response

Response to Reviewer 2 Comments

I reviewed the manuscript “Medical Students’ Perspective on LGBTIQ+ Health Care Education in Germany” for the IJERPH.

The research topic is interesting, but the approach is punctual and generalizations are limited.

Point 1: Title: Please make your title more accurate as far as findings are concerned and may maximize the chances of reach and citation; however, its main finding remains unclear. The use of questions is also interesting for the title.

Response 1: We appreciate the reviewer’s suggestion and edited our title as follows:

Medical Students’ Perspectives on LGBTQI+ Healthcare and Education in Germany: Results of a Nationwide Online Survey 

Point 2: Abstract: Suffers from the same problem as the title. I highlight the following problems:

  1. It looks like it was written for professionals in the research area, and if so, it's less likely to be read. Understand your summary as a continuation of the title. The originality must be very clear in the abstract, and its conclusion must necessarily respond to the objective, respecting the adopted method; Structure in objectives, methods, results and main implications. The results must be highlighted

Results should present the key results, and not be tied to methodological details. In conclusion rather than repeating the results, authors should focus on answering the objectives thinking about the PRACTICAL IMPLICATIONS of their findings. In other words, why is your text interesting? What should an “average/non-expert” reader learn from your text? Highlight this in the conclusion.

Use the first person for this (we suggest that / we perceive that...). It is not possible to see the authors in the manuscript and this is a little disappointing.

Response 2: Thank you for the constructive comments. Based on this feedback, we edited and rewrote the abstract substantially to highlight our research’s novelty, clearly summarize our study’s objective and its findings, and highlight central conclusions. Please note that we structured the abstract according to IJERPH recommendations, thus without the headings suggested by the reviewer:

The healthcare needs of lesbian, gay, bisexual, trans*, queer, and intersex (LGBTQI+) persons are often overlooked, prompting national and international calls to include diversity-related competencies into medical students’ training. However, LGBTQI+ focused healthcare education targets remain elusive, as surveys reveal considerable variability across national student populations. To generate empirical data and vocalize recommendations for medical education, we conducted the first nationwide online survey among 670 German medical students from 33 universities. Overall, most respondents reported low confidence regarding their medical training preparing them for LGBTQI+ patients, stated that LGBTQI+ themes were not covered during training, and agreed that the inclusion of such themes is urgently needed. In addition, we found gender and LGBTQI+ community member status to be key variables. Men scored lower in knowledge than women, while community members scored higher than non-community members. Similarly, community members reported higher comfort levels. Non-community men showed the highest levels of prejudice and efficacy beliefs, while at the same time had the lowest scores in contacts and perceived importance of LGBTQI+ related teaching. Keeping subgroup differences in mind, we recommend that educational trainings should include LGBTQI+ healthcare aspects and address self-efficacy beliefs in future medical professionals to overcome LGBTQI+ healthcare disparities.

Point 3: Introduction:

The Introduction contains primary errors and fails to state the rationale for the research. It only limits itself to pointing out what other authors have already found, without differentiating, or pointing out where it is advancing.

In general, it needs to contextualize the theme nationally and internationally, there is no highlight for the knowledge gap reinforced by the literature.

It must necessarily POINTING AND PRESENTING the knowledge gap that it came to fill (HIGHLY RECOMMENDED); it is not enough to say that it exists. Remember that the gap must be auditable, so it is necessary to carry out a systematic search or point to a current (integrative or systematic) review;

To say that “There is a gap in knowledge...” is vague.

Response 3: We acknowledge that the introduction required improvement, and thank the reviewer for the helpful suggstions. We have edited and extended the introduction therefore substantially as part of our revision. Specifically, we now contextualize our research topic as concerning fundamental human rights:

Recent polls estimate that one in every 14 US Americans and central Europeans (around 7%) identifies as lesbian, gay, bisexual, trans*, queer, or intersex (LGBTQI+) [1,2]. Despite increasing international and national efforts towards respecting their fundamental rights [3], LGBTQI+ persons face continuing discrimination [4–7], including in healthcare settings [8–10].

We delineate the rationale of our study by pointing towards recent systematic reviews that highlight the importance of advancing medical students’ education and the critical role of student perspectives:

Physicians’ prejudice against LGBTQI+ persons, a lack of knowledge of LGBTQI+ relevant healthcare needs, and low comfort levels and self-efficacy beliefs in working with LGBTQI+ patients are frequently noted as potential sources of healthcare discrimination [23,24], prompting national and international calls for including diversity-related competencies into medical students’ training [25,26]. In a systematic review, Morris et al. recently identified 13 LGBTQ+ focused educational programs for healthcare students and providers, concluding that specific interventions show promise for improving LGBTQI+ related attitudes, community- and health-related knowledge, and comfort levels in interacting with LGBTQI+ patients [24]. Cautioning against universal approaches, the authors further note the importance of accounting for medical students’ perspectives when planning educational interventions. Students' awareness of health disparities and motivation for change may critically determine the success of bias-related trainings. Moreover, student perspectives may help identify relevant targets and preferred educational strategies among the plethora of available options [23], which appears crucial given the still low inclusion rates of LGBTQI+ content in medical curricula [27–29].

We then explain how our survey advances research on student perspectives, which thus far remains inconsistent:

Thus far, students’ perspectives on LGBTQI+ healthcare and education remain barely explored, with an informal search of existing surveys revealing considerable variability between different student populations. Students surveyed at a US American medical school expressed high levels of comfort in interacting with LGBTQI+ patients, but felt unprepared by their formal education [30]. At other US American medical schools, medical students agreed on the importance of sexual history taking with LGBTQI+ patients, although they also expressed lower comfort levels when compared to other patient populations [31]. Medical students surveyed at a UK university expressed positive attitudes towards LGBTQI+ healthcare education as well as a need for improved training, with low confidence in using and clarifying unfamiliar sexual and gender terms emerging as a particular concern [32]. Similar results emerged in another UK survey, revealing low confidence levels in discussing patients´ gender identity independent of the number of years of medical training [33]. Surveys among German and Hungarian students further revealed that knowledge of and attitudes towards LGBTQI+ focused medical education may depend on student sociodemographic features, for example, their own gender [34] or their religiosity [35]. This suggests that LGBTQI+ healthcare education may require tailoring at both national and subgroup levels. 

Finally, we clearly state our study’s objective, while acknowledging the selectivity of our approach:

Given that information regarding medical students’ perspectives on LGBTQI+ healthcare and education remains sparse, we conducted the first nationwide online survey among medical students enrolled at several German medical faculties. Specifically, we assessed students’ LGBTQI+ related knowledge, prejudice, comfort levels, and their experience with and perspectives on LGBTQI+ healthcare and education. The goal of the present study was to generate empirical data and to vocalize recommendations regarding adaptations in medical education and trainings for future generations of physicians in Germany, to meet LGBTQI+ healthcare needs. In addition, we considered students´ gender (men vs. women) and LGBTQI+ community status (member of the LGBTQI+ community vs. non-member) as relevant variables in our analyses. Although recommendations derived from a national survey remain selective, we nevertheless expect that advancing this line of research contributes toward the overall goal of tackling discrimination and improving the situation of LGBTQI+ persons in healthcare settings.

Point 4: Method

There is a lot of information needed to analyze the data. Authors should follow the strobe checklist to guide the method: https://www.strobe-statement.org/checklists/

Response 4: In revising the methods section, we have followed STROBE recommendations as far as applicable to our survey. Thus, we now include sections on “Study Design, Setting, and Participants”, and “Recruitment”. We also extended the description of the “Procedure” and “Statistical Analysis” to conform with recommendations as best as possible. Please refer to the main manuscript for changes.

Point 5: Data analysis is unclear and does not respect the principles of manuscripts that employ quantitative analysis.

Response 5: We have specified the “Data Analysis” section to provide additional information. Please refer to the main manuscript for changes.

Overall survey responses were analyzed descriptively using means, standard deviations, and percentages. Univariate analysis of variance (ANOVA) was used to compare responses between respondent subgroups as defined by gender and LGBTQI+ community status. Tukey’s post hoc comparisons were used to follow up on significant interaction effects. Regression analyses were used to explore associations between potential targets of LGBTQI+ focused medical education (i.e., knowledge, prejudice, comfort, and efficacy beliefs) and students’ averaged teaching and private experiences. The significance level for all inferential analyses was set at p ≤ 0.05. No formal power analysis was conducted given the absence of prespecified inferential hypotheses. All analyses were performed with R in RStudio (2020; R version 3.5.6; RStudio version 1.1.463).

We also made substantial edits to the description of the results to improve our manuscript´s overall clarity and presentation.

In order to identify potential group differences among survey variables, respondent subgroups based on gender (men vs. women) and LGBTQI+ community status (community member vs. non- member) were compared using separate 2 (Gender) x 2 (Community Status) ANOVAs. Means, SDs and inferential statistics are summarized in Table A1. N = 11 diverse respondents were excluded from analyses due to the small group size, but group means were nevertheless included in the table for the sake of completeness. Post hoc comparisons are summarized in Table A2.

Knowledge levels differed based on gender and community status, with men showing lower knowledge levels than women, p = .026, and members showing higher knowledge levels than non-members, p < .001. The interaction was not significant, p = .318.

Prejudice levels differed by gender and community status, and were qualified by a gender x community interaction, ps < 001. Non-community men scored significantly higher than non-community women, p < .001, community women, p < .001, and community men, p < .001. Other comparisons were not significant, ps > .43.

Contact levels differed by community status, p < .001, and were qualified by a gender x community interaction, p = .01. Non-community men reported fewer contact than non-community women, p = .006, community women, p < .001, and community men, p < .001. Non-community women reported fewer contact than community women, p < .001, and community men, p < .001. Community men and community women did not differ in reported contact levels, p = .93.

Comfort levels differed only by community status, p < .001, but not by gender or their interaction, ps > .58; community members reported higher comfort levels than non-members.

Efficacy beliefs varied by gender, p < .001, and were qualified by a gender x community interaction, p < .001. The main effect of community status was not significant, p = .10. Post hoc comparisons revealed that non-community men scored higher than non-community women, p < .001, community women, p < .001, and community men, p = .001. All other comparisons were not significant, ps > .99.

Reported levels of experienced teaching did not differ based on gender, community status, or their interaction, all ps > .26. Perceived importance of teaching, however, differed by gender and community status, and were qualified by a gender x community interaction, ps < .001. Non-community men reported lower levels of perceived importance than non-community women, p < .001, community women, p < .001, and community men, p < .001. Non-community women also scored lower than community women, p = .003. Differences between community men and women, and between non-community women and community men were not significant, ps > .06.

Point 6: The same applies to the results and tables, which must make clear the tests used.

Response 6: As noted above, we have edited our results section substantially to improve its overall clarity and accuracy in presentation. Used tests are now clearly stated at the appropriate positions in the text as well as in the tables. For example, in the subgroup comaprisons, we now clarifiy:

In order to identify potential group differences among survey variables, respondent subgroups based on gender (men vs. women) and LGBTQI+ community status (community member vs. non- member) were compared using separate 2 (Gender) x 2 (Community Status) between-participants ANOVAs. Means, SDs and inferential statistics are summarized in Table A1. N = 11 diverse respondents were excluded from analyses due to the small group size, but group means were nevertheless included in the table for the sake of completeness. Post hoc comparisons are summarized in Table A2.

Point 7: What are the inclusion and exclusion criteria?

Response 7: Please refer to section 2.1 and 3.1, where we define that:

All students enrolled in any medical faculty at a German university were eligible for the study.

and

Thirteen individuals reported not to be studying medicine and were therefore excluded from analysis.

Point 8: What is the origin of the seeds? How was it publicized? Was there dissemination among peers through social networks?

Response 8: Please refer to section 2.2, which describes the recruitment process (changes compared to our previous manuscript version are highlighted):

Respondents were recruited using a convenience sampling method, recruiting as many respondents as possible within a five-month window through emails from the working group Sexuality and Prevention of the German Medical Students' Association (bvmd e.V.). The working group´s national coordination distributed links using a nation-wide email distribution list reaching 1052 German medical students directly. The group further appealed to local representatives and sexual education project groups to distribute the survey via additional local email distribution lists. We also advertised the survey in medical lectures held at the Ruhr-University Bochum, Campus East-Westphalia. The recruitment period was August to December 2021.

Point 9: What are the allocation criteria? Was there a sample calculation? Was there a power test of the sample?

Response 9: We now specify in section 2.2 (see above) that we used a convenience sampling approach. As we further add in section 2.5. (Data Analysis, see above), no formal power analysis was conducted given the absence of prespecified inferential hypotheses. The survey had no independent conditions or allocation criteria.

Point 10: What are the platform's security criteria?

Response 10: In the Procedure (Section 2.4), we have now added the following:

The online survey was hosted on a university-supplied, password- and SSL-secured webserver and implemented using the JavaScript-based open-source platform jsPsych [36].

Point 11: And what about the questionnaire? Was there validation? What is the process of creating it? Was there a pre-test? How and by whom?

Response 11: Please refer to Section 2.3, in which we have now specified the survey development.

A 57-item survey was developed in a multistage process involving an expert panel of psychologists, physicians, experts in trans* health, and members of the Workgroup Sexuality and Prevention of the German Medical Students' Association (Bundesvertretung der Medizinstudierenden in Deutschland (bvmd) e.V.). The panel generated an initial set of 117 items, which were evaluated for validity, objectivity, and comprehensibility, and subsequently reduced to a final set of 57 items (see Supplementary Materials).

Point 12: Discussion:

The authors repeat some results in the discussion rather than explaining their findings. The results seem to continue and this is an error. You should focus on explaining/validating your original findings. What should the average reader take away from your discussion? Do not write to the specialist as he is able to understand your text.

Don't focus on just pointing out what the literature brings and justifying the similarities. For what reasons do they exist? Why are there differences? Explain, and validate your findings.

Remember to use arguments that validate your results.

This will require a readaptation of the discussion.

Response 12: We appreciate the general suggestion to further substantiate and explain our findings in terms most readers would understand. We have therefore made several edits and changes within the discussion, including omitting repetitions of our findings, and adding explanations of our findings instead. Given the numerous changes throughout the text, please refer to the highlighted sections within the revised manuscript for further details.

Point 13: Study limitations must be DISCUSSED. How were the limitations circumvented so as not to influence the results?

Response 13: We extended the discussion of limitations accordingly, and now also explain how we addressed potential limitations that we anticipated a priori. However, addressing some limitations may require further research, which we discuss accordingly:

Although conducted with a large sample of medical students from a range of German universities and addressing the role of respondent characteristics in subgroup comparisons, we cannot exclude that self-selection biases may have occurred in our study. About 30% of respondents self-identified as LGBTQI+ community members compared to prevalence estimates of 7.4% in Germany [2]. We suspect that the distribution mode may have promoted LGBTQI+ participation as invitation recipients of the particular student groups were likely familiar and already engaged with the subject. As a result, our findings could overestimate the medical student population’s knowledge and positive attitudes. However, as even individuals familiar with the subject expressed low belief in their training to prepare them for encountering LGBTQI+ patients in an adequate manner, calls for addressing student comfort and efficacy beliefs are still warranted.

We must further note potential limitations regarding our survey. Although developed by an expert panel to ensure content validity, the survey has not been evaluted in terms of its internal structure and psychometric properties. For example, we assessed LGBTQI+-related knowledge on a general level rather than for each subgroup individually, which previous surveys suggest may have revealed lower knowledge levels among specific subject areas  (e.g., trans*-health, [49]). Further psychometric evaluation is therefore required. Besides the structure of student knowledge, such future research should address different dimensions of prejudice, comfort, efficacy beliefs, and evaluate their relevance for predicting student interactions with LGBTQI+ patients. Indeed, Morris et al. [24] noted that measures used in evaluating educational interventions have been rarely examined in terms of their consequences for behavior, which should be remedied in future research.

Point 14: Conclusion

The conclusion is very generic and similar to other sections. There is repetition. It must answer the objectives, be written in the first person and bring THE RESEARCHER'S VIEW ON THE FINDINGS. The conclusion is personal; that is, it is the look of you of the findings. Think about what your work changes, and why. It should not be focused on technical terms but on the implications.

Use direct, blunt language to explain what your findings were.

Response 14: We re-wrote the conclusion to focus on our findings´ implications for whether and how LGBTQI+ related healthcare should be included in medical curricula. Specifically, we now write:

Medical students in Germany agree on the importance of LGBTQI+ related health care education but require further training. We found gender and LGBTQI+ community member status to be key variables within the context of LGBTQI+ related medical training. Our results reveal that, overall, men had lower scores in knowledge than women, while community members had higher scores than non-community members. Similarly, community members reported higher scores in comfort levels. Finally, non-community men scored higher in prejudice, lower in contacts, higher in self-efficacy beliefs, and lower in perceived importance of teaching than all other groups (non-community women, community men, community women). With these findings in mind, we recommend that educational trainings should include LGBTQI+ healthcare aspects and address self-efficacy beliefs in future medical professionals to overcome LGBTQI+ healthcare disparities.
